# Green Synthesis and Applications of ZnO and TiO_2_ Nanostructures

**DOI:** 10.3390/molecules26082236

**Published:** 2021-04-13

**Authors:** Rosana A. Gonçalves, Rosimara P. Toledo, Nirav Joshi, Olivia M. Berengue

**Affiliations:** 1Department of Physics, School of Engineering, São Paulo State University (UNESP), Guaratinguetá, Sao Paulo 12516-410, Brazil; rosana703@gmail.com (R.A.G.); rosimara.passos@gmail.com (R.P.T.); 2São Carlos Institute of Physics, University of São Paulo, 369, São Carlos, Sao Paulo 13560-970, Brazil

**Keywords:** metal oxide nanostructures, green synthesis, ZnO, TiO_2_, industrial applications

## Abstract

Over the last two decades, oxide nanostructures have been continuously evaluated and used in many technological applications. The advancement of the controlled synthesis approach to design desired morphology is a fundamental key to the discipline of material science and nanotechnology. These nanostructures can be prepared via different physical and chemical methods; however, a green and ecofriendly synthesis approach is a promising way to produce these nanostructures with desired properties with less risk of hazardous chemicals. In this regard, ZnO and TiO_2_ nanostructures are prominent candidates for various applications. Moreover, they are more efficient, non-toxic, and cost-effective. This review mainly focuses on the recent state-of-the-art advancements in the green synthesis approach for ZnO and TiO_2_ nanostructures and their applications. The first section summarizes the green synthesis approach to synthesize ZnO and TiO_2_ nanostructures via different routes such as solvothermal, hydrothermal, co-precipitation, and sol-gel using biological systems that are based on the principles of green chemistry. The second section demonstrates the application of ZnO and TiO_2_ nanostructures. The review also discusses the problems and future perspectives of green synthesis methods and the related issues posed and overlooked by the scientific community on the green approach to nanostructure oxides.

## 1. Introduction

Green technologies have gained enormous attention over the last decade. Natural resources are being depleted daily, and the green approach appears to be a prominent solution without destroying natural resources. This technology deals with the fabrication of nanomaterials and their applications in the medical, sensor, optoelectronics, energy, food industries, etc. [1]. There are many physical and chemical methods of preparing metal nanoparticles (NPs) and metal oxide nanostructures such as sputtering, lithography, and electrospinning. However, they are quite expensive, and involvement with toxic chemicals results in health risks. In this regard, the green synthesis approach does not require any harmful chemicals, high-pressure reactors, or high temperatures. Most importantly, it results in degradable waste with less risk of contamination at the end [2,3]. Over the past few years, researchers have shown interest in green chemistry to synthesize NPs using environmentally benign agents such as plants, fruits, flowers, algae, yeasts, bacteria, fungi. Additionally, extensive research has been carried out using plant extracts for the synthesis of NPs, and it was observed that compared to other means, plants are more suitable for the production of NPs, even at the pilot scale [4,5,6,7,8,9].

Nanostructured semiconductor metal oxides are a class of materials that play an important role in the development of most electronic devices such as solar cells, transistors, diodes, and sensors [10,11,12,13,14,15,16,17,18,19,20,21,22,23,24,25,26]. Among various metal oxides, ZnO and TiO_2_ NPs are of much interest in the scientific community due to the fact of their unique electronic, chemical, and physical properties, their high surface-to-volume ratio, and availability of more surface atoms for an immediate chemical reaction [11,14,27,28]. Zinc oxide and titanium dioxide are *n*-type wide bandgap semiconductors (Eg = 3.37 and 3.6 eV, at 300 K, respectively) [9,29]. These compounds have drawn the interest of many researchers due to the wide range of their application. They are highly acceptable for commercialization due to the fact of their shape, size, conductivity, etc. Meanwhile, depending on their morphology, these materials can be prepared via various top-down and bottom-up methods. However, the green synthesis approach has motivated researchers to achieve the desired properties, size, and shape because of their facile one-step approach and environmentally friendly protocol. Nevertheless, some factors are always kept in mind while performing green synthesis, such as pH, reaction temperature and time, stability, risk assessment, and regulatory challenges [30,31]. Figure 1 shows the schematic illustration of green synthesis approaches and applications that we explore in the following section, along with the progress in the scientific articles on green synthesis to synthesize ZnO and TiO_2_ nanostructures in recent years.

## 2. Green Synthesis Methods of TiO_2_ and ZnO Nanostructures

This review summarizes the most widely used green synthesis methods in the fabrication of TiO_2_ and ZnO nanostructures for technological applications such as photocatalytic, supercapacitor, solar cell, and gas sensors. Our main objective was to shed light on the scope of solvothermal, hydrothermal, co-precipitation, and sol-gel methods and their advantages, drawbacks, and research advancements.

### 2.1. Sol-Gel Synthesis

Sol-gel is a widely used method of synthesizing ceramic oxide nanostructures from solution by transforming liquid precursors to “sol” and ultimately to a network structure called “gel” in wet chemical phases [32]. The composition of the sol is usually achieved by hydrolysis and condensation of metal alkoxide precursors. Still, a sol may generally be called a colloidal suspension involving a more comprehensive range of systems. There are also several different ways to form a gel, which Flory described in 1974. He divided gel into four groups: lamellar gel, ordered gel, disordered particulate gels, and physically aggregated polymers [33]. Later in 1996, Kakihana demonstrated more classifications of different gels [34]. Table 1 outlines the five major categories of gel types used in sol-gel chemistry. Since there are many articles on the sol-gel synthesis approach used to fabricate various nanostructures, our goal was to show the green approach using sol-gel chemistry.

Recently, researchers have demonstrated a high interest in the development of metal oxide NPs through a greener approach because they are eco-friendly, less toxic, and generate less hazardous waste. In this context, nanostructured oxide semiconductors have generated considerable interest due to the fact of their fundamental importance in addressing some of the main issues in fundamental physics and their possible applications as advanced materials. These nanostructures have been prepared by a variety of different fabrication methods. However, the reproducible and scalable synthesis of these nanostructures is the major difficulty for technological applications. Furthermore, many of these methods require expensive equipment and have little control over the scale, shape, and composition of NPs. In this way, the green approach of sol-gel processes can overcome some of the major limitations.

#### 2.1.1. Green Sol-Gel Synthesis Approach for ZnO Nanostructures

Tu Uyen Doan Thi et al. [36] utilized the green sol-gel method to produce ZnO NPs by orange fruit peel extracts and studied the impact of pH and temperature on the morphology and antibacterial activities. Figure 2a shows a schematic of the ZnO NPs’ synthesis in which zinc nitrate and orange extract powder were mixed, followed by annealing at 400 °C for one hour. Figure 2b shows the reaction mechanism between the orange peel extract and zinc precursor, where the orange peel extract acts as a ligand agent. They analyzed the morphology and antibacterial properties by annealing the ZnO NPs at different temperatures ranging from 300–900 °C. Figure 2c shows the X-ray diffraction pattern (XRD) of the ZnO NPs annealed at a different temperature; it can be seen that compared to high-temperature samples, the crystal structure and diffraction peaks were poor at a lower temperature. By increasing the annealing temperature, the crystalline size increased, and NPs became reoriented and reduced the number of defects in grain boundaries. Figure 2d shows the Fourier transform infrared spectra (FTIR) of ZnO NPs under different annealing temperatures; a vibration bonding Zn–O and orange peel extract at 450 and 1640 cm^−1^ were observed, respectively.

Moreover, at low annealing temperatures, residual organic extract vibrations in the NPs were present, which gradually disappeared at higher annealing temperatures. Thermal stability and weight loss of the ZnO samples were evaluated by thermogravimetry Analysis (TGA). Significant weight loss was observed at a lower annealing temperature due to the loss of moisture and organic substances. In contrast, no weight loss was observed at higher annealing temperatures (Figure 2e).

Similarly, Sasirekha et al. [37] fabricated the ZnO/C nanocomposite via a cost-effective sol-gel method via a green approach in which they used sucrose as a capping agent. They studied the structural and electrochemical behavior of the prepared ZnO/C. Moreover, they performed electrochemical measurements which revealed a maximum specific capacitance of 820 F g^−1^ with a current density of 1 A g^−1^. Moreover, when they performed charge–discharge up to 400 cycles, a power retention of 92% was observed. In another work by Silva and co-workers [38], it was reported for the first time the facile green sol-gel synthesis of ZnO NPs using whey as a chelating agent, characterizing the samples using different analytical techniques.

Sahoo and co-workers [39] reported ZnO NPs using acacia concinna fruit extract as a surfactant. They used the *acacia concinna* powder to prepare the Zinc precursor solution, followed by a mixing and calcination process to achieve pure ZnO NPs. Figure 3a shows the XRD pattern of pure ZnO NPs, which reveals the hexagonal wurtzite phase. The ZnO sample’s sharp peak indicates the crystalline nature of the materials, and the Debye–Scherer equation measured the crystalline size (26 nm). Structural analysis of the ZnO sample was confirmed by scanning electron microscopy (SEM), and it can be seen that all the NPs were agglomerated (Figure 3b,c). The UV-Vis spectra of the ZnO NPs were analyzed at 498 nm at different time intervals, and it was found that there was a steady decrease in the intensity of the Congo red (CR) dye over time (Figure 3d). The FTIR analysis was carried out for the ZnO NPs, CR dye, and CR-loaded ZnO NPs (Figure 3e). The peaks observed at 1386 cm^−1^ and 1681 cm^−1^ indicate the C=O group’s asymmetric stretching vibration. In addition, the peak at 1057 cm^−1^ was due to the occurrence of m (C–N), and the peaks at 1235 and 1178 cm^−1^ were due to the aromatic C–N stretching and absorption of CR. Even after loading the CR with ZnO NPs, there were no changes in peak positions and bands. Figure 3f displays the transmission electron microscopy (TEM) images of ZnO NPs. It appears that the particles were irregular with an average particle size of approximately 30 to 50 nm. Similarly, the research group of Guy Van Assche [40] reported the cost-effective green synthetic route “sol-gel injection” to incorporate ZnO NPs onto porous silica matrix. They also confirmed that by enclosing ZnO NPs in the silica matrix, there is a way to monitor the size of the particles, the size distribution, and the NPs’ ability to aggregate and open up a new possibility to explore the application for catalysis and optical detection.

Majid Darroudi et al. [41] studied the temperature effect of zinc oxide NPs prepared using gum tragacanth (GT)—a green, economical, and readily available polysaccharide component. They mixed zinc nitrate as a zinc source and GT in the water and fixed it at 80 °C in an oil bath. The final product was washed, cleaned, and calcined at different temperatures (400–700 °C). Figure 4a displays the powder XRD of the ZnO NPs at different calcination temperatures. It can be seen that all the diffraction peaks with miller indices were indexed to a pure ZnO wurtzite structure with a crystalline size below 50 nm. The UV-Vis spectra of the ZnO NPs showed a 3.3 eV bandgap, and a sharp absorption peak at 370 nm was observed, which can be assigned to the absorption of the intrinsic bandgap because of the transition from the valence band to the conduction band (Figure 4b). The TEM images (Figure 4c,d) indicate the narrow size distribution of the ZnO NPs in GT media with a mean size of approximately 33 nm confirmed with Image J software. These NPs are expected to find potential applications in various fields such as cosmetics, paints and coatings, sensors, and medicines.

Similarly, Araujo et al. [42] reported a novel approach to preparing ZnO NPs using karaya gum—a polysaccharide extracted from low-cost *Sterculia* species. The prepared ZnO NPs were characterized by analytical techniques and their photocatalytic performance was studied. Moreover, some biosynthesis approaches for the synthesis of ZnO nanostructures were reported using plant extracts, fruit, and leaves (Table 2).

#### 2.1.2. Green Sol-Gel Synthesis Approach for TiO_2_ Nanostructures

Several synthetic approaches to the preparation of TiO_2_ nanostructures have been developed to date. The sol-gel method’s green approach is widely used because of the eco-friendly and cost-effective concept for obtaining TiO_2_ nanostructures. Li Yang et al. [89] reported the sol-gel route to synthesize TiO_2_ microtubes using *Platanus acerifolia* seed fibers (SFs). Figure 5a,b show SEM images of the natural seed fibers that display the hollow tubular structures with a diameter of 25–30 μm and a wall thickness of 3–5 μm. Figure 5c,d show the titanium sol-impregnated fibers (TIFs), and Figure 5e shows the titanium fiber hybrid (TFH). The TiO_2_ microtubes were obtained with further calcination of the TFH at 500 °C for two hours with an average diameter of 24 μm and a wall thickness of 2 μm (Figure 5f–h). With the high magnification of the image with the TiO_2_ microtubes, it can be seen that the wall of the microtubes consisted of two layers, where NPs covered the outer wall with a rough surface, and the inner wall was very smooth and compact (Figure 5i).

Furthermore, Figure 5j shows the enlarged view of the TiO_2_ microtubes’ inner wall that was composed of a dense structure of TiO_2_ nanospheres of a diameter of 25 ± 5 nm. We can further understand the double soaking sol-gel route of the synthesized TiO_2_ microtubes based on the morphological changes. Figure 5k shows a schematic of the fabrication process of the TiO_2_ microtubes, and the first soaking step was to hydrolyze natural seed fibers with titanium sol to generate the metal alkoxide layer on the fibers’ surface. Once the Titanium sol was completely hydrolyzed on the deionized water, they formed the rough external wall that was the second soaking step. The final product was obtained with the calcination step to obtained anatase TiO_2_ microtubes. The XRD pattern of the TiO_2_ microcubes and the microtubes’ internal wall (Figure 5l) were analyzed and confirmed that all the peaks were indexed to the pure anatase TiO_2_ phase. There was a slight shift towards a lower degree in the XRD pattern of the internal wall (TiO_2_ microcubes) because of fine TiO_2_ NPs, which we had seen in the SEM image. Figure 5m shows the FTIR spectra of the natural SFs and TiO_2_ microtubes, TIF, and TFH prepared from natural SFs. The major characteristics peaks of C=O, C=C, and C–OH at 2937, 1735, and 1033 cm^−1^ from cellulose, hemicelluloses, and lignin, while the enlarged view of flexural vibration of O–H at 1630 cm^−1^ was attributed due to the existence of the moisture in the *Platanus acerifolia* seed fibers. Figure 5n shows the surface area obtained by the Brunauer–Emmett–Teller (BET) method and pore distribution of the TiO_2_ microtubes. A mesoporous size distribution was observed with a surface area of 128.2 m^2^/g with a pore diameter of 3.553 nm, which is higher than that reported for the commercial P25 TiO_2_. The UV-Vis spectra of synthesized TiO_2_ microtubes and commercial P25 TiO_2_ were analyzed (Figure 5o). The sharp absorption edge for TiO_2_ microtubes and commercial P25 TiO_2_ microtubes were observed at 400 and 390 nm. The bandgaps were measured and were 2.81 eV and 3.17 eV for TiO_2_ microtubes and commercial P25 TiO_2_, respectively; this difference was observed due to the carbon in the densely packed TiO_2_ NPs. This research demonstrated the use of a novel double soaking sol-gel route to synthesize TiO_2_ microtubes that have two advantages: they are environmentally friendly and green and they have excellent properties.

In another work, Muhammad Atif Irshad and co-workers reported a novel sol-gel approach using plant extracts (*Trianthema portulacastrum* (T2) and *Chenopodium quinoa* (T3)) to synthesize TiO_2_ NPs. They also compared this green method with a chemical process to analyze the antifungal activity and observed that TiO_2_ prepared via the green approach showed a better antifungal response against wheat rust. The various green sources have been used for TiO_2_–NPs synthesis via the sol-gel route, as reported in Table 3, which gives a broader view of the sol-gel method’s green approach.

Despite the beneficial aspects of TiO_2_ NPs, a few minor drawbacks limit the practicality of its application. The bandgap of anatase TiO_2_ is 3.23 eV, which can hamper the photocatalyst’s efficiency due to the fast recombination rate of the generated charge carriers, combined with a slow transfer rate of electrons to oxygen. Therefore, doping or modification of TiO_2_ NPs with noble metals, such as gold, silver, platinum, and palladium, is an alternative way to lower the bandgap and to promote and enhance visible light absorption. Hariharan et al. [90] reported the fabrication of Pd@TiO_2_ NPs using *Aloe vera* gel, which acts as a capping and reducing agent during fabrication. In another work, Rostami-Vartooni et al. [91] reported a novel sol-gel approach to fabricating Ag–TiO_2_ nanocomposites using *C. acinaciformis* leaf and flower extracts to achieve the desired photocatalytic properties. They confirmed that silver-doped-TiO_2_ nanocomposites showed faster photocatalytic degradation than pure TiO_2_ NPs. 

### 2.2. Co-Precipitation Method 

Metal oxides can be synthesized using the co-precipitation method through a two-step process: first, the precipitation of metal hydroxides and, second, a heat treatment to crystallize the oxide. In the homogeneous liquid phase, the nucleation and growth kinetics of the particles are determined by the controlled release of anions and cations in the solution, and the shape and size distribution can be adjusted by controlling parameters such as pH and concentration of reagents/ions [99]. The co-precipitation method has some advantages over other chemical routes such as low cost, low energy and time consumption, and the possibility of large-scale production [100,101]. In addition, green synthesis can be easily adapted to this method including TiO_2_ and ZnO nanostructures. 

Mainly, green co-precipitation synthesis involves the use of plant extracts (leaf, root, fruit, bark). The biocomponents present in the extract can act as a stabilizing, reducing, capping, or chelating agent, changing both the morphology and the properties of the nanostructures, improving their performance in applications. Other strategies for obtaining nanostructures through green routes include the use of less aggressive solvents, fewer reagents, less energy consumption, the use of renewable feedstocks, reduced derivatives, and production of self-degrading products [102]. In general, the green syntheses of TiO_2_/ZnO nanostructures produced by the plant extract-mediated co-precipitation method follows an approximately similar route: First, a precursor solution is mixed with a solution of the plant extract under agitation. Then, changes in the temperature and/or pH of the mixture (addition of NaOH) initiates the precipitation process and hydroxide formation. The third step consists of the separation (simple or by centrifugation) of the precipitate, washing with deionized water and/or ethanol accompanied or not by a drying process at low temperatures. The last step consists of thermal treatment (calcination) at higher temperatures to crystallize the oxide. The final product can also be ground in a crystal mortar pestle. Figure 6 depicts a schematic diagram containing all these steps for a generic green synthesis using the co-precipitation method. 

#### 2.2.1. Green Co-Precipitation Synthesis Approach for TiO_2_ Nanostructures

In general, reports of the green syntheses of TiO_2_ nanostructures using the co-precipitation method in the literature are still modest but have shown notable growth in recent years due to the method’s ease of use and timesaving qualities. Next, we report on some works in this area. In recent years, Rawat and collaborators have produced different TiO_2_ nanostructures using a green co-precipitation method. They synthesized spherical TiO_2_ NPs (20–30 nm) using *Phyllanthus emblica* (Amla) leaf extract and TTIP as a titanium source [103]. In a typical process, the TiO_2_ precursor and the leaf extract (1:1 ratio by volume) were mixed and stirred at room temperature for 20 min until the color of the solution changed from transparent to whitish brown. Then, ammonia was added drop by drop to the solution, initiating the formation of the precipitate. Anatase NPs were obtained after filtering the solution and rinsing the precipitate with alcohol, calcining in a muffle furnace (400 °C, 3 h), and grinding in a crystal mortar pestle. This same method was previously used to obtain anatase nanocubes using *Tinospora cordifolia* leaf extract [104]. In this, it was observed that the control of the morphology provided by the extract’s biomolecules proved to be fundamental in the performance of the nanostructures for degradation of the neutral red (NR) dye: cubic TiO_2_ was more efficient for dye photodegradation than non-cubic NPs, reaching a percentage of 93.4% against 65.3% after 120 min under UV illumination.

Many other works have reported using plant extract mediating the synthesis of anatase TiO_2_ NPs using the co-precipitation method [105,106,107,108]. However, as with traditional syntheses, green syntheses of the rutile phase are scarcer. One of the disadvantages of the co-precipitation method is the rapid nucleation and growth of the nanostructures; the accelerated growth can result in a strong agglomeration of the final product. However, many green syntheses have shown that using plant extracts can improve the dispersion of the synthesized NPs using this method. Goutam et al. [95], for example, synthesized TiO_2_ NPs using *Jatropha curcas* L. leaf extract and showed that the low formation of NP agglomerates was related to the performance of phytochemicals present in the extract as capping agents. A similar effect was observed by Subhapriya and Gomathipriya [109] and by Kaur et al. [94] in the green synthesis of polydisperse TiO_2_ NPs from *Trigonella foenum* leaf extract and *Lagenaria siceraria* leaf extract, respectively. 

In this scenario, we highlight the green co-precipitation synthesis in which no plant extract was used, as reported by Muniandy et al. [110], for the production of mesoporous anatase NPs using TTIP as a titanium source, water as a solvent, and starch as a template. According to the authors, starch plays a key role in the formation of mesoporous structures: the nucleation and initial crystal growth occur when the precursor diffuses and forms complexes with amylose molecules close to the interspaces between the swollen starch microspheres. With the calcination process, these starch granule templates are removed, giving rise to mesoporous structures. The morphology of the synthesized structures in this work was analyzed by Field Emission scanning Electron Microscopy (FESEM) and High-Resolution Transmission Electron Microscopy (HRTEM) and can be seen in Figure 7. The influence of the amount of Ti precursor and the pH of the solution on the morphology and photocatalytic activity of the material was also evaluated in this work.

#### 2.2.2. Green Co-Precipitation Synthesis Approach for ZnO Nanostructures

As for the production of ZnO nanostructures, there is a more abundant number of papers using the co-precipitation method, especially when this process is mediated by plant extract. This is largely due to the work of Singh et al. [56], who reported a simple and efficient method for green synthesis of ZnO NPs using this method. In the mentioned work, zinc oxide NPs were produced using latex from *Calotropis procera* as an alternative to chemical syntheses. For the synthesis, 0.02 M of aqueous zinc acetate dihydrate solution was mixed with 50 mL of distilled water with vigorous stirring. After 10 min, 0.25, 0.5, and 1.0 mL of latex were added, one part at a time, to the acetate solution. Then, NaOH (2.0 M) was added to the solution until it reached pH 12, then the obtained mixture was placed on a magnetic stirrer for 2 h. The precipitate was collected, rinsed repeatedly with distilled water and ethanol to remove impurities, and left to dry overnight in a vacuum at 60 °C. The final product was spherical NPs (5–40 nm) and granular nanosized with little agglomeration; it remained stable and without visible changes even one month after synthesis. These characteristics were attributed to the effective role of latex as a stabilizing and reducing agent. From this pioneering work, many other green syntheses mediated by plant extracts have been reported for the production of zinc oxide NPs using the co-precipitation method. Table 4 presents works in the literature reporting the use of this type of green synthesis to produce ZnO nanostructures and the functionality of each of the biocomponents used in its preparation.

Regarding the syntheses that do not use plant extract for the synthesis of ZnO nanostructures, we can highlight the work of Akir et al. [125] in which structures with different morphologies were produced from three different protocols for the addition of basic solution in the zinc aqueous solution: spherical NPs, nanosheets, and hexagonal prismatic NPs, indicating that the speed of addition of the basic solution to the zinc precursor is a key factor for the morphology of the structures. Charoenthai and Yomma [126] synthesized ZnO NPs with a hexagonal wurtzite structure using a process similar to that adopted by Akir et al. [125]. In this case, the authors showed that the use of water as a solvent, in comparison with ethanol, methanol, propanol, and butanol solvents, results in the formation of smaller NPs with greater surface area and greater pore volume that caused an increased photodegradation for methylene blue (MB) and methyl orange (MO) dyes.

### 2.3. Hydrothermal Method

Nanoparticle synthesis is widely studied today, and many processing methods have been developed to produce homogeneous structures with high crystalline quality. Among all methods used to synthesize nanostructures, hydrothermal synthesis has been considered one of the most promising methods in this regard. In this method, the stability provided by using a closed system, where temperature and pressure are controlled, allows greater control over the size, nucleation, and degree of crystallinity of the NPs [127,128]. Thus, the hydrothermal method has been used to synthesize several nanostructures, including TiO_2_ and ZnO nanostructures (mono-dispersed and highly homogeneous NPs, nano-hybrid materials, and nanocomposites), as part of the framework of green synthesis [99]. In general, green hydrothermal syntheses are synthesized by non-toxic solvents and non-corrosive solutions in their process, which minimizes damage to the environment and reduces the consumption of raw materials. There are still few reports on green synthesized TiO_2_ and ZnO nanostructures compared to other methods such as co-precipitation and sol-gel. However, promising results have been presented in the literature and are summarized as follows.

#### 2.3.1. Green Hydrothermal Synthesis for ZnO Nanostructures

Recently, Chang et al. [129] used a green hydrothermal route for the synthesis of versatile nanostructured zinc oxide particles (nZnOs) from zinc acetate precursors (Figure 8a). The morphological characterization showed that the NPs were shaped like nanorods (Figure 8b), nanoplatelets (Figure 8c), and multibranched flower-like particles (Figure 8d) for growth times of 6, 12, and 24 h, respectively. Figure 8e shows the nZnO nanostructure used in antimicrobial activity testing. The multibranched flower-type ZnO presented more remarkable, reliable, and stable antifungal activity than the other nZnOs, probably because it has a larger surface area.

Guo et al. [130] showed that the reaction temperature and time in the hydrothermal method were also fundamental to defining the crystalline phase of the material and the morphology. Changing the autoclave treatments at 100–120 °C for 6 h (or 170 °C for 0 h) to treatment at 170 °C for 3–6 h, the obtained product changed from cubic ZnO_2_ nanocrystals to hexagonal ZnO nanorods. The UV-Vis absorption spectra showed the synthesized ZnO_2_ nanocrystals had optical bandgaps around 4.1 eV, and the ZnO nanorods presented at 3.3 eV, which makes both structures good candidates for applications in photocatalysis and optoelectronic devices with a short wavelength. In another work, Lam et al. [77] proposed a green hydrothermal approach for the large-scale synthesis of ZnO nanotubes (NTs) using powder ZnO and aqueous H_2_O_2_ solution (volume 30%) as starting materials and performing an autoclave treatment at 130 °C for one day. The ZnO NTs with an average diameter of 10 nm and a wall thickness of 3–5 nm (rolling the nanosheet layer) were applied as photocatalysts for degradation of endocrine chemical disruptor methylparaben under UV irradiation. The degradation of methylparaben has been associated with the unique tubular structure and the large surface area of the NTs of ZnO that give rise to increased separation of electrons and holes and the formation of a large number of reactive radicals in the photocatalytic process. Liu et al. [131] also used the same method with an aqueous solution of H_2_O_2_ as a solvent for the production of hollow ZnO NTs and nanospheres (treatment in an autoclave at 120 °C for 6 h). An interesting study was also presented by Patrinoiu et al. [111] regarding control over ZnO nanostructures’ morphology; different nanostructures were produced by the variation in concentrations of zinc acetate precursor and starch reagent in the hydrothermal synthesis (autoclave treatment at 180 °C for 24 h). The authors claimed that the key parameter for the morphological alterations was the gelation capacity of amylose released by the starch. The study also showed that all the ZnO nanostructures exhibited antibacterial activity and antibiofilm potential.

#### 2.3.2. Green Hydrothermal Synthesis for TiO_2_ Nanostructures

Wang et al. [132] successfully synthesized TiO_2_ NPs with different morphologies (i.e., nanorods, nanospheres, and microspheres) and crystalline phases by variating oxalic acid (OA) and TTIP precursor concentrations in a surfactant-free green hydrothermal approach. At first, OA was dissolved into H_2_O and magnetically stirred until a transparent solution was obtained. The TTIP was dropped until a colorless solution was reached. The solution was hydrothermally treated at 180 °C for 12 h. Deposits were collected by vacuum filtration, washed in deionized water and anhydrous ethanol, and calcinated at 80 °C for 12 h in a vacuum box. To reach different TiO_2_ phases (i.e., anatase and rutile), several TTIP/OA molar ratios were used, from 2:1 to 1:1, 1:3, 1:6, and 1:9 with the same initial route. Their work also showed that the microspheres with mixed anatase and rutile phases presented better photocatalytic performance.

Similarly, Spada et al. [133] used annealing temperature variations of 400–1000 °C (4 h) to control the crystalline phase and the particle size of TiO_2_ NPs. The XRD data show that at temperatures above 600 °C, the anatase phase started a transition to the rutile phase, and at temperatures above 1000 °C pure rutile was found. The size of the crystals increased proportionally with the increase in temperature, ranging from 17 to 57 nm; on the other hand, the bandgap decreased from 3.21 to 2.93 eV. Degeneration tests of the rhodamine B (RhB) dye showed that the NPs obtained at 600 °C, with small fractions of the rutile phase, presented improved photocatalytic performance. Green hydrothermal synthesis mediated by plant extracts were also used for the production of TiO_2_ NPs. Recently, Hariharan et al. [134] synthesized TiO_2_ NPs using *Aloe vera* gel and deionized water as starting materials (autoclave treatment at 180 °C for 4 h). Sample characterization showed TiO_2_ anatase NPs with good crystalline quality and sizes ranging from 6 to 13 nm. In addition, the use of plant extract resulted in a better performance of NPs for picric acid photodegradation than NPs synthesized by the chemical hydrothermal route. Subsequently, Hariharan et al. [135] used the same method for producing Ag-doped TiO_2_ NPs. The Ag–TiO_2_ NPs were found to have an improved activity for photodegradation of picric acid under visible light and also showed anticancer activity, decreasing the growth of cancer cells and increasing the reactive oxygen species.

### 2.4. Solvothermal Method

Solvothermal synthesis is a widely used process to synthesize several technological materials such as ferrite [136,137,138], Sn: In_2_O_3_ [139,140], CeO_2_ [141,142], ZnO [143,144,145,146,147], Metal Organic Frameworks (MOFs) [148,149,150,151,152], TiO_2_ [153,154,155,156,157,158]. In this approach, an organic solvent (non-aqueous) is used as reactional media in which a solute is dissolved and, subsequently, crystallized under high-temperature and high-pressure conditions. These conditions are achieved by confining the solution to a special sealed system such as an autoclave. There are several advantages associated with using the solvothermal approach instead of other inorganic synthesis, but by far the most attractive one is the simplicity of the approach. Other significant advantages are the control over shape, size distribution, and crystalline phases. Despite the tremendous success of the solvothermal approach for synthesizing TiO_2_ and ZnO nanostructures [143,144,145,146,147,153,154,155,156,157,158], it is still an approach that uses highly corrosive and toxic chemical precursors for human beings. For this reason, it is essential to seek greener approaches to synthesize these metal oxide nanostructures, since they are of high technological interest. In this context, our goal was to present the most recent bibliography on the green solvothermal approach to synthesize TiO_2_ and ZnO NPs.

#### 2.4.1. Green Solvothermal Synthesis Approach for ZnO Nanostructures

In this section, the green solvothermal approach is presented in a slightly broader concept by considering papers that use plant extracts and natural templates and references in which authors tried to simplify their methodologies, reducing the quantity of precursors, or substituting hazards solvents or not using surfactants. In general, there are still few reports on the green synthesis of ZnO and TiO_2_ nanostructures in which the solvothermal approach was employed. Despite this fact, Zhang et al. [159] successfully synthesized Er–Al co-doped ZnO NPs by using a one-pot and surfactant-free, non-toxic solvothermal approach. The authors characterized the samples’ structural features and photocatalytic activity by degrading methyl orange (MO) in water under visible light irradiation. The XRD, energy-dispersive X-ray spectroscopy (EDX), and X-ray photoelectron spectroscopy (XPS) measurements revealed that Er and Al ions doped the ZnO’s pristine structure. The UV-Vis measurements revealed that the co-doping caused a widening in the ZnO bandgap from 3.14 eV (i.e., pristine structure) to 2.95 (Er–Al co-doped sample) and, consequently, an increase in the visible light absorption of ZnO. The authors also showed that Er–Al co-doped ZnO presented an enhanced photocatalytic activity under visible light illumination with 98.9% MO degradation efficiency. Šutka et al. [160] showed a straightforward and ethanol-based solvothermal synthesis of plasmonic Al-doped ZnO (AZO) NPs using Al and Zn salts as starting materials.

The control of ZnO samples was synthesized by mixing zinc acetate dehydrate (Zn(CH_3_CO_2_)_2_∙2H_2_O) in ethanol, resulting in a 0.1 M solution be mixed with a second solution prepared with NaOH in ethanol. Previously to their mixing, both solutions were vigorously stirred and heated to 80 °C. After this, the two solutions were mixed and left under stirring at 80 °C for 10 h. The final mixture was then transferred into a 50 mL Teflon-lined, stainless-steel autoclave, sealed, and heated at 150 °C for 24 h. The resulting material was filtered and washed with methanol and dried in air at 60 °C for 5 h. Doped samples were synthesized by substituting an amount of Zn(CH_3_CO_2_)_2_∙2H_2_O in ethanol by Al(NO_3_)_3_∙9H_2_O. Structural characterization showed that the Al atoms effectively substituted Zn in the zincite structure. Morphological features of the undoped and doped ZnO samples were studied by SEM and TEM measurements as presented in Figure 9.

Figure 9a,c show that the ZnO samples have a nanowire morphology with diameters ranging from 20 to 70 nm, lengths ranging from 0.2 μm to 1 μm, and aspect ratios up to 50. The increase of the Al dopant in the ZnO structure (Figure 9b,d) causes the aspect ratio to decrease, leading the final products to be NPs and nanorods (lengths below 50 nm and diameters around 10 nm). The authors showed via diffuse reflectance measurements the presence of localized surface plasmon absorption in the NIR region for all doped samples. They demonstrated the doped samples could be used for technological applications such as a hyper-realistic piezoresistive sensor based on a composite material formed by mixing Zn_0.925_Al_0.075_O nanocrystals into polydimethylsiloxane. In another work, Liu and co-workers [161] reported the synthesis of an octahedral ZnO/ZnFe_2_O_4_ heterostructure through a surfactant-free solvothermal method followed by thermal treatment. At the first step, the precursor to the ZnO/ZnFe_2_O_4_ final product was synthesized using the solvothermal method in which ZnCl_2_ and FeCl_3_·6H_2_O were dissolved in ethylene glycol (EG), and to this mixture, it was added CH_3_COONa. This mixture was then stirred for 30 min and then sealed into a Teflon-lined, stainless-steel autoclave (200 °C for 12 h). The final octahedral ZnO/ZnFe_2_O_4_ was achieved after the precursor was annealed in air at 500 °C. The authors showed the water treatment promising character of octahedral ZnO/ZnFe_2_O_4_ samples because of their excellent adsorption capacity of malachite green (MG) and selectivity in mixtures of dyes such as in MG/MO and MG/RhB. Mahlaule-Glory et al. [162] synthesized ZnO NPs using an eco-friendly approach to the traditional solvothermal method in which *Sutherlandia frutescens* extract was used as a reducing and capping agent for the synthesis of ZnO NPs. The plant composite formed by the *Sutherlandia frutescens* and ZnO NPs showed bactericidal effects against Gram-positive and -negative strains and antiproliferative effects against the A549 human alveolar lung cancer cell line. Although the authors claim that they used the solvothermal approach to synthesize the ZnO nanostructures, the parameters used in the synthesis were not described in the text.

#### 2.4.2. Green Solvothermal Synthesis Approach for TiO_2_ Nanostructures

Several authors reported the use of titanium alkoxide as the metal precursor of solvothermal green synthesis of titanium oxide nano- and microstructures [163,164,165,166]. Wang and co-workers [164] reported a successful, single-step, green synthesis of the monoclinic metastable phase of TiO_2_ known as TiO_2_ (B). The samples’ structural characterizations confirmed the crystalline phase, and their composition was mainly formed by titanium and oxide atoms. The HRTEM measurements revealed that the as-synthesized TiO_2_ (B) comprised tiny crystallites and nanoporous structures (Figure 10a,b). In Figure 10c,d, the fast Fourier transition (FFT) image revealed the crystalline character of the sample, and the inverse fast Fourier transition (IFFT) image showed the (0 0 1) plane of TiO_2_ (B), respectively. Wang et al. [164] also found that green synthesized TiO_2_ (B) was highly efficient and stable for the decomposition of MO dye in agreement with previously reported studies in which non-green synthesized TiO_2_ (B) was found to be a highly efficient phase for the degradation of this dye. 

Additionally, TiO_2_ core–shell microspheres were synthesized by a template-free and hydrofluoric acid-free solvothermal synthesis starting from TTIP, isopropyl alcohol, and organic amine [166]. The photocatalytic activity was also studied under visible irradiation and UV-Vis irradiation. Structural characterization data showed that the TiO_2_ core–shell was mainly composed of NPs aggregates covered by perpendicular assembled nanosheets with high-energy {116} facets exposed. The XPS measurements revealed that in situ doping with nitrogen at the interstitial sites of TiO_2_ shells occurred and induced local states above the valence band edge, leading to the narrowing of the bandgap and resulting in a visible light response of the material. Zhao et al. [167] reported the synthesis of spinous hollow pure anatase TiO_2_ and ZrO_2_-doped TiO_2_ microspheres using a solvothermal green approach in which sunflower pollen acted as bio templates. Shortly, the methodology used for the TiO_2_ pure phase was based on the dispersion of sunflower pollen template in absolute ethanol and then the addition of titanium butoxide under continuous magnetic stirring. After 2 h, water was added, and the mixture was submitted to the solvothermal conditions. A calcination process was also conducted after the solvothermal synthesis to ensure the samples’ crystalline and stoichiometry. Doped samples were synthesized in a very similar way by mixing titanium butoxide and zirconium *n*-butoxide instead of titanium butoxide.

In Figure 11a, the sunflower pollen morphology was close to a sphere covered with spines at the surface. Without calcination, ZrO_2_-doped TiO_2_ spinous hollow microspheres presented a very similar morphology as depicted in Figure 11b. The micrographs obtained for pure TiO_2_ and doped samples synthesized with 4.6%, 8.8%, 12.6%, and 18.2% molar ratio of zirconium *n*-butoxide in mixed esters can be seen in Figure 11c–f, respectively. These results indicated that all samples retain the spherical shape of the pollen templated besides the different amounts of ZrO_2_ introduced in the synthesis. Calcined samples presented smaller diameters due to the removal of the pollen templates (Figure 11g), and the hollow structure is presented in Figure 11h by observation of a broken microsphere in the micrograph. The authors claim that the samples’ hollow features probably be originated from the release of CO_2_ during the carbonization process of organic matter in pollen. Figure 11i depicts the samples synthesized without templates. Zhou and collaborators [168] reported the synthesis of anatase TiO_2_ mesocrystals using a green solvothermal method based on a halide precursor TiCl_3_. The authors showed that the as-synthesized samples were mainly composed of anatase mesocrystals with the Wulff construction in which the facet exposed was {101}.

## 3. Applications of TiO_2_ and ZnO Nanostructures

### 3.1. Gas Sensor Applications

The living standards of the human race in the 20th century grew rapidly due to the industrial revolution. Industrialization demands specific gas detection and monitoring for the benefit of society [16,18,169,170,171,172,173]. These include hydrocarbons (for the exploration of oil fields), oxygen (for breathable atmospheres and combustion processes, e.g., in boilers and internal combustion engines), and other various gaseous chemicals (for medical applications, manufacturing of different chemicals, etc.). However, extensive industrialization has a negative aspect: the emission of polluting gases into the environment poses a risk to public health. Therefore, gas sensors need to measure pollution in the atmosphere to take adequate control measures [13,17,21,174].

Due to the fact of their unique optical, electrical, and chemical properties, semiconductor metal oxide (SMO) nanomaterials, such as SnO_2_, ZnO, and TiO_2_, have created high expectations as sensitive layers. Moreover, SMO-based resistive gas sensors have been extensively used because of their low cost, compact size, and easy production. The SMO’s sensing properties depend on their morphology and the type and concentration of defects generated during the synthesis and doping. As discussed earlier, there are many techniques to fabricate ZnO and TiO_2_ nanomaterials to control their morphology. Moreover, recent studies show that various nanostructures, such as nanowires, nanocubes, and nanobelts, have enhanced sensor response to toxic gases because of the high surface-to-volume ratio. However, selectivity and high operation temperature hamper the use of these sensors for commercial purposes. With green synthesis, we can achieve the same morphology as other techniques, since they are eco-friendly, energy efficient, and take less time to process than different approaches. However, reproducibility and mass production of NPs via the green approach still requires more research.

The ZnO NPs prepared from *Aloe vera* plant extract [26] were tested for gas sensing properties and compared with the chemical method. Those NPs were observed to show maximum sensitivity towards 1000 ppm of liquified petroleum gas (LPG) at 250 °C, and it was concluded that both techniques showed similar responses towards LPG; however, green synthesized nanomaterials are recommended due to the fact of their facile approach. The ZnO nanocubes were synthesized using alginate, a water-soluble polysaccharide and a desirable candidate for aqueous processing; it showed room temperature selective sensing towards ammonia gas [175]. Besides their advantages, one of the most challenging issues for SMO-based gas sensors is to achieve selectivity towards target analytes under controlled humidity. Previous works to improve the specificity of metal oxides include: (i) incorporating suitable additives, (ii) temperature control, and (iii) using appropriate filters. A recent study showed that selectivity and sensitivity could be further enhanced by doping some oxides/2D materials, decorating the surface with noble metals, and using UV-light illumination [13,174,176,177,178,179,180]. In summary, nanomaterials prepared via a green approach could help to achieve high sensitivity; however, selectivity and operation temperature are still a challenge with metal oxide, and there are still many experiments and discoveries required to overcome this issue.

### 3.2. Photocatalysis Applications

Photocatalysis is usually defined as the process in which a photoinduced reaction is accelerated by a catalyst material. When irradiated with photons, the catalyst produces electron–hole pairs that can interact with other molecules in the reaction medium, giving rise to reactive oxidative species (ROS) that can degrade toxic components in less dangerous species. Traditionally, semiconductor materials have been used as a catalyst due to the fact of their unique electronic structure, the bandgap between the full valence band and the empty conduction band is low enough to allow these materials to be sensitizers for light-induced redox processes. In this category, ZnO and TiO_2_ oxides also stand out for their low cost, chemical inertness, thermal stability, and low hazard. [181]. In the particular case of nanostructures produced by green syntheses, the main photocatalytic application of TiO_2_ and ZnO oxides is in the treatment of wastewater, with the decolorization of water contaminated by dyes being the primary target. Outstanding works that report the use of these nanostructures for the degradation of dyes are listed in Table 5.

Notably, most of the photodegradation processes in Table 5 are associated with RhB, MB, and MG; this is because they are the primary dyes found in effluents [206]. In green syntheses, in which ZnO and TiO_2_ nanostructures are produced by means of plant extract, the degradation of harmful agents can be aided by the oxidizing potential of biocomponents such as quinones, phenols, and flavonoids present in the extracts; reduction potential will depend on the plant species, the type of dye analyzed and the temperature [207]. As highlighted in Table 5, NPs obtained with the aid of plant extracts presented excellent catalytic performances for dye degradation (synthesis marked with an asterisk) comparable to that of NPs obtained by other types of green syntheses and also chemical syntheses. Despite the excellent performances, the phenomenon of reduction and the role of biocomponents in photodegradation have been little discussed and need further investigation. Some studies have suggested that these biocomponents can help to generate more hydroxyl radicals on the semiconductor surface, resulting in an increase in photocatalytic activity [185,188].

More recently, in addition to the usual applications in dye photodegradation, TiO_2_ NPs synthesized by green approaches have also been used to remove Cr^6+^ ions and chemical oxygen demand of real tannery effluents (efficiency of 82.26% and 76.48% under solar illumination) [95], the photoreduction of Cr^6+^ ions (79.6% under UV irradiation) [208], removal of Pb from explosive industrial wastewater (82.53% removal after a 12 h treatment with UV light) [108], removal of total organic carbon and total nitrogen in refinery wastewater (efficiency of 32% and 67% under UV-C lighting, respectively) [209], and also organic compounds such as picric acid (100% under visible light at 120 min) [134], ornidazole antibiotic (67% efficiency under UV illumination after 100 min) [210], and ciprofloxacin (CIP) antibiotic (CIP removal of 90% after 60 min under UV-Vis illumination) [211]. The photodegradation of phenol (80% and 100% under irradiation with visible and UV light, respectively) [212], anthracene (96% efficiency a 4 h treatment with UV illumination) [213], photocatalytic generation of H_2_ (360 µmol/g under UV-Vis irradiation) [214], and photocatalytic treatment against *Enterococcus faecalis* bacteria (99.2% efficiency under visible illumination) [118] was also achieved by using ZnO nanostructures produced by green synthesis. In general, the photocatalysis process can be affected by factors such as surface defects that can both reduce the bandgap, decreasing the energy needed to produce photoexcited carriers [198,212,215], act in the prevention of the recombination of carriers [110], or increase the production of •OH radicals [84,203]; texture effects [53]; morphology, by an increase of surface area and/or active sites; and, in this case of ZnO, to its absorption capacity in a wide range of the solar spectrum [193]. The improvement in these aspects constitutes the best way to tailor the use of nanostructures produced by green routes in photocatalytic processes, and efforts should be made in this regard. Although the biggest challenge in the area is to show a superior performance of the ZnO and TiO_2_ catalysts for applications other than the well-established dye photodegradation and to elucidate the role of biocomponents (derived from plant extracts used in most green synthesis) in the photocatalytic process using them in their interest.

### 3.3. Supercapacitor Application

Over the last decades, many researches have focused on developing high-performance supercapacitors with greater storage capacity, faster loading, high-temperature resistance, and low cost. Metal oxides have been widely used as electrode materials in supercapacitors because of their high specific capacitance and low resistance, which allow the construction of high-energy devices. Currently, there has been an increase in the interest in producing oxide nanostructures for the development of new supercapacitors by using green approaches in which the environmental impacts can be reduced without impairing the good properties of the electrodes [216,217]. Recently, Reddy et al. [217] synthesized high surface area TiO_2_ NPs from the *Ocimum tenuiflorum* (OT) extracts, and Calotropis gigantea (CG) plants for the development of electrodes. The synthesized NPs showed a high specific capacitance of 105 F·g^−1^ for the OT and 224 F·g^−1^ for the CG compared with conventional TiO_2_-based electrodes, which indicates that green synthesized TiO_2_ NPs are efficient for electrochemical energy storage devices. Dhanemozhi and co-authors [218] reported ZnO NPs synthesis using the *Camellia sinensis* plant extracts (i.e., green tea) and evaluated their capacitance features as potential candidates to the development of supercapacitor applications. The ZnO NPs demonstrated excellent CV characteristics and good electrochemical stability, indicating that the as-prepared material can be used for supercapacitors applications. Similar results were achieved by Anand et al. [219] and Lee et al. [220] on ZnO nanostructures synthesized with the aid of *Prunus dulcis* and *Chlorella vulgaris* plant extracts, respectively.

As in traditional synthesis, strategies such as functionalization with metallic NPs and the formation of composites have also been used to improve the capacitance of NPs produced by green approaches. Aravinda et al. [221] showed that ZnO decorated carbon NTs nanocomposite electrodes presented a significant increase in specific capacitance and good stored energy density than pure NTs electrodes. Rajangam et al. [222] observed a similar behavior from Ag decorated TiO_2_ NPs synthesized using rose petals. Despite the reported significant advance in this area, many challenges need to be overcome to produce high-quality devices such as durability, electrode stability, high-capacity retention, good cycling stability. Although the use of oxide nanostructures itself has been proving that such challenges can be overcome, improvements are still needed in their synthesis approaches to develop active electrodes in which imbalance at redox sites, degradation, rise in internal resistance, and increase in equivalent series resistance can be minimized. In this scenario, the green synthesis approach can be the solution to these issues with the great advantage of low cost since the precursors are abundant in nature [223,224].

### 3.4. Solar Cell Application

Solar cells are an important way to produce clean and renewable energy, given their abundance and continuous availability. Over the last decades, research in this area has expanded due to the high global energy demand and the effects caused to the environment and climate by using fossil fuels. The ZnO and TiO_2_ nanostructures synthesized by green approaches have been used in the manufacture of third-generation solar cells (comprising emerging technologies not yet available on the market), especially of Dye-sensitized Solar Cells (DSSCs), a subclass of thin-film solar cells that has shown to be a promising alternative to silicon solar cells in view of the low cost, efficiency and easy manufacturing. In this device, the process of converting sunlight into electricity is based on the sensitization of a wide bandgap semiconductor used as photoanode material [102,225].

The TiO_2_ semiconductor is commonly used in DSSC photoanodes due to the fact of its properties such as its small particle size, high surface area, highly active anatase phase, high bandgap energy, low density, and high electron mobility [102,226], in addition to its non-toxicity, easy availability, and low cost. However, the ZnO semiconductor has also gained attention as a photoanode material. Characteristics, such as a wide bandgap, the high exciton binding energy (60 meV), strong luminescence, high thermal conductivity, and greater electron mobility than TiO_2_, have driven these studies [225,227]. Despite the advantages of combining the use of the TiO_2_ and ZnO semiconductors produced by green synthesis with the manufacture of solar cells, studies in this area are still scarce, although promising. Deng et al. [201] recently found a very expressive result when manufacturing a DSSC photoanode depositing mesoporous spheres of rutile TiO_2_ produced by green hydrothermal synthesis on FTO substrate coated with a dense layer of TiO_2_ anatase NPs and sensitizing them with the dye N719. Due to the greater light capture and the high specific surface area provided by the mesoporous spheres, the device achieved a conversion efficiency of 8.43%, 18% higher than that found for the reference DSSC (single layer of NPs). Ullattil and Periyat et al. [228] reached a slightly lower performance using anatase mesoporous NPs synthesized by a green microwave method as photoanode material in a DSSC (6.58% conversion efficiency). However, in general, the conversion efficiency achieved in DSSCs has been more modest with values changing from 2.79% to 4.33% for DSSC with photoanodes based on anatase NPs biosynthesized [229,230,231], a 3.8% efficiency for a DSSC based on mixed-phase anatase and rutile TiO_2_ nanorods synthesized from *Phellinus linteus mushroom* extract [232], an efficiency between 0.63–2.10% for DSSCs with ZnO NPs-based photoanodes [114,233,234,235].

Strategies to improve the performance of the DSSCs photoanodes have evolved doping of titania with Zn^2+^ ions [236] (efficiency increases from 4.4% to 4.8%), morphology changes [227], use of quantum dots instead of dye as sensibilization agent [237], use of a polymer-based electrolyte instead of the traditional liquid electrolyte in DSSCs (5.2% efficiency, TiO_2_ NPs) [226], and quasi-solid DSSCs (5.50–6.46% efficiency, ZnO nanosheets, and building blocks) and formation of composites between ZnO and natural graphite (3.12% efficiency) [238], biosynthesized ZnO, commercial TiO_2_ and graphene oxide (4.61–6.18% efficiency) [239]. In addition to DSSCs, other types of solar cells have also been developed using TiO_2_ and ZnO nanostructures synthesized by green routes, the records include the manufacture of polymeric solar cells (PSCs) with the addition of TiO_2_ NPs coating (25% higher efficiency than is found for uncoated PSCs) [240], the fabrication of hole-conductor-free perovskite solar cells from mesoporous TiO_2_ NPs (8.52% efficiency, 21% higher than that found for commercial TiO_2_ based devices) [163] and Pitchaiya et al. [241] in the manufacture of perovskite solar cells in bilayer using ZnO nanostructures between a TiO_2_ NPs film and the perovskite layer (7.83% efficiency). The challenges in the development of third-generation solar cells based on nanostructures involve not only (i) the improvement of conversion efficiency, well below the limit reached by solar cells based on monocrystalline silicon, through the processing of new materials/components or improvement of existing ones; but also (ii) reliability and lifetime, devices must offer long-term stability and good resistance to moisture, heat, and impact; (iii) large-scale manufacturing, based on the implementation of new production techniques with critical dimensional control resources, structural homogeneity and higher yield that allow taking advantage of the quantum resources of nanostructures; (iv) reduction in manufacturing costs [242,243,244]. The use of TiO_2_ and ZnO nanostructures synthesized by green routes, although it may not solve the first three problems, is a good bet for reducing production costs through more environmentally friendly processes with less energy consumption, time, and raw material.

### 3.5. Photocatalytic Water Splitting Application

The conversion of solar energy into electrochemical energy as fuel from water splitting has emerged as an efficient and low-cost strategy for clean and renewable energy production. Hydrogen (H_2_) produced in this photocatalytic process is considered one of the most promising green fuels for the future due to its high energy-per-mass content, zero CO_2_ emission, low-cost, operation facility and capability of separating H_2_ and O_2_ streams [245,246,247].

In photoelectrochemical cells (PEC), H_2_O can be converted directly to H_2_ under solar irradiation, using a simple photocatalytic process as follows: (i) photogeneration of electron-hole pairs in the photoanode (in general, semiconductor oxide materials); (ii) separation and transfer of the photogenerated electron-hole pairs to the electrolyte and counter-electrode; (iii) reaction of the evolution of O_2_ (water splitting) on the surface of the photoanode by combining the photogenerated and H_2_O holes simultaneously with the evolution of H_2_ in the counter electrode from the combination of the H^+^ ions and the photogenerated electrons [248,249,250,251,252]. This water splitting reaction requires a minimum redox potential of 1.23 eV, so the semiconductor used with photoanode material must absorb photons with energy greater than 1.23 eV for the process to be viable and above 2.0 eV for the rate of reaction is satisfactory [253]. More detailed information on the water-splitting process can be found at [254,255,256].

The TiO_2_ and ZnO semiconductors obtained by green syntheses are particularly interesting for use as photoanode materials in the H_2_ generation process due to the simple, inexpensive and eco-friendly method of producing the material [257]. Also, the robustness, abundance, and non-toxicity contribute to the choice of these semiconductors as photocatalysts [258]. Both ZnO and TiO_2_ have been considered promising photocatalysts, both are capable of producing photogenerated holes with high oxidizing power. The efficiency of H_2_ generation with the use of these photocatalysts is limited by the wide bandgap and the high presence of electron-hole recombination centers; and, in the case of ZnO, due to the easy dissolution in aqueous solution under UV irradiation. Considering that the photocatalytic activity is strongly affected by the size [259], shape [258], and defects of the photocatalysts [260], the optimization of the morphology and crystalline structure [261,262] has been studied, and several micro-and nanostructures of ZnO and TiO_2_ have been presented.

Recently, for example, Hu et al. [263] showed enhanced photocatalytic activity for TiO_2_ anatase nanoplates produced by green solid-state synthesis. Nanoplates with exposed facet (001) and with a high degree of crystallinity present a generation rate of H_2_ (13 mmol h^−1^ g^−1^) and current density (0.22 mA cm^−2^), higher than the obtained for other structures reported in the literature from conventional syntheses. In addition, the study shows that the photocatalytic generation of H_2_ is strongly dependent on the sample’s crystallinity and texture effects. Similar studies have also demonstrated good performance of fibrous hierarchical meso-macroporous N-doped TiO_2_ (364.3 µmol/h) [264], TiO_2_/Pt NPs nanocomposites (4.92 mmol h^−1^ g^−1^) [265] and carbon-modified TiO_2_ composite (21 mL h^−1^ g^−1^) [266], synthesized by green routes for the production of H_2_ via water splitting. Archana et al. [214] prepared ZnO nanoparticles by a green combustion method and used them as a photocatalyst for the photocatalytic generation of H_2_. The water-splitting experiments showed smaller nanoparticles have a higher H_2_ evolution rate, reaching a maximum of 360 µmol/g under optimized conditions. This enhanced efficiency was associated with an abundance of oxygen vacancies in the sample. Also, the study shows that no photo-corrosion process was observed during the reaction, indicating the excellent photostability of the NPs; it is believed that the residual carbon of the synthesis is responsible for this. Promising photochemical water splitting results have also been achieved using Ag-ZnO NPs biosynthesized as a photocatalyst; under ideal conditions, the evolution rate of H_2_ reaches 214 mmol g^−1^ h^−1^ [267].

Despite the advantages and promising results obtained for the photocatalytic generation of H_2_ by water splitting on PECs, this technology still faces many technical difficulties that limit its insertion in the market: (i) the hydrogen and oxygen generated must be immediately separated for reasons of safety, system complexity and product yield (without reverse reaction), which requires additional energy consumption, reducing the global efficiency of the process; (ii) development of more efficient mechanisms for capture, separation, storage and purity of H_2_ gas (highly explosive); (iii) development of high quality photoanodes capable of rapid separation/extraction/transport/injection of carriers with low rate of recombination, ability to effectively absorb visible light and strong ability to capture incident photons in order to increase efficiency of the reaction of H_2_ generation (~18% currently); current PEC cells need changes in materials and design to acts as electrochemical reactors and for large-scale production [256,268,269,270,271,272,273]. Although slowly, efforts by the scientific community have been moving in this direction.

## 4. Conclusions and Future Perspective

The green synthesis approach of oxide nanostructures has been the area of focused research for the past few years. Green sources, such as plants, flowers, and bacteria, are acting as stabilizing and reducing agents to control nanomaterials’ morphology. This review has provided a review of recent advancements, the challenges in green synthesis routes, and the suitability for TiO_2_ and ZnO nanostructures for advanced applications.

Herein, firstly, the types of green approaches to synthesize ZnO and TiO_2_ nanostructures based on traditional and widely used synthesis methods, such as hydrothermal, solvothermal, sol-gel, and co-precipitation, are highlighted. The possibility of merging low-cost and control of parameters, such as shape and size, with eco-friendly synthesis is the major justification for the employment of future efforts in developing a sustainable and scalable production of nanostructures. Different green sources, such as plant extract, bacteria, flower, and algae, and their use in these synthesis methods show marked variation in morphologies, surface area, porosity, and properties. The major advantages were discussed regarding green approaches such as the use of bio precursors for the synthesis, low waste of chemicals, reduced toxicity, replacement or suppression of the use of hazard solvents, low-energy waste for synthesis or chemical processes. Moreover, the effect of reaction parameters, such as pH, temperature, and reaction time, were discussed. Secondly, the applications of ZnO and TiO_2_ have been widely explored already in previous review articles; therefore, we summarized the most important technological applications with their challenges, strengths, and future perspectives. As in gas sensors, we discussed how we can overcome problems such as operating temperature and selectivity issues. Similarly, we discussed challenges related to efficiency, manufacturing costs, lifetime, and large-scale production of solar cells based on nanostructures. In discussing photocatalysis, we showed the challenges in understanding the role of biocomponents used in syntheses and to optimize the structure and morphology of nanostructures to develop new applications in the area. As for supercapacitors, we presented some of the problems involved in their manufacture, such as durability, electrode stability, high-capacity retention, and toxicity, and how the use of greener nanostructures can solve some of them.

From future perspectives, one of the central challenges starting from now would be the development of new green methods to synthesize ZnO and TiO_2_ NPs and the improvement of the already existing methods with the optimization of reactions to, consequently, achieve an improved quality of the as-synthesized products. The extrapolation of laboratory experiments to an industrial scale will be important, too, as new green solutions to synthesize nanomaterials will be developed, especially those based on bio precursors, which have rapid change in their physical and chemical properties. The nanostructured materials prepared via green synthesis have a huge application in biomedicine, pharmaceutical, and food industries and, thus, will become a major research area in the next few years.

## Figures and Tables

**Figure 1 molecules-26-02236-f001:**
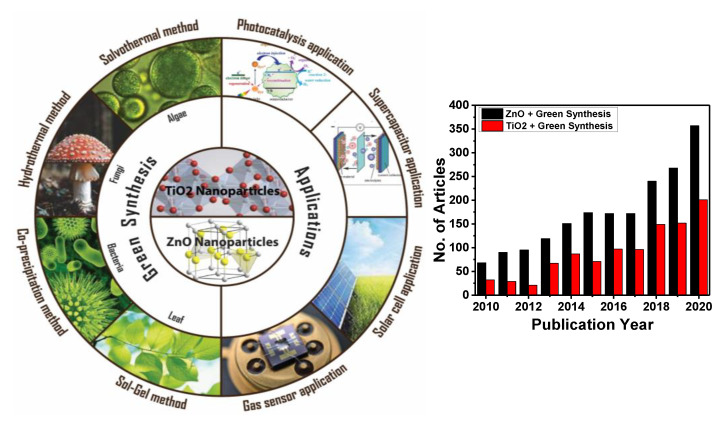
Schematic illustration of green synthesis approach of ZnO and TiO_2_ nanoparticles (NPs) and their application. The number of publications on green synthesis of ZnO and TiO_2_ NPs from 2010 to 2020 (internet search of Scopus on 10 March 2021). Keywords searched: ZnO + Green Synthesis and TiO_2_ + Green Synthesis.

**Figure 2 molecules-26-02236-f002:**
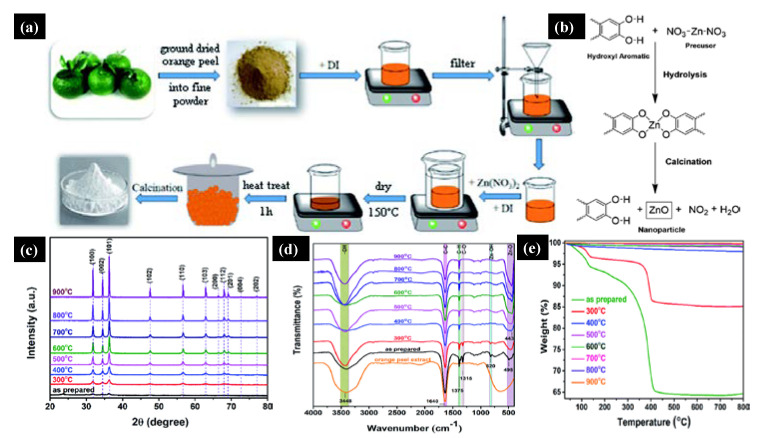
(**a**) Schematic of the green sol-gel synthesis of ZnO NPs. (**b**) Chemical mechanism of the ZnO NPs’ formation. As synthesized and annealed ZnO samples at temperatures of 300–900 °C. (**c**) X-ray diffraction pattern. (**d**) FTIR Plot. (**e**) Thermogravimetric spectra. Figure adapted with permission from Reference [36]. Copyright 2020 RSC.

**Figure 3 molecules-26-02236-f003:**
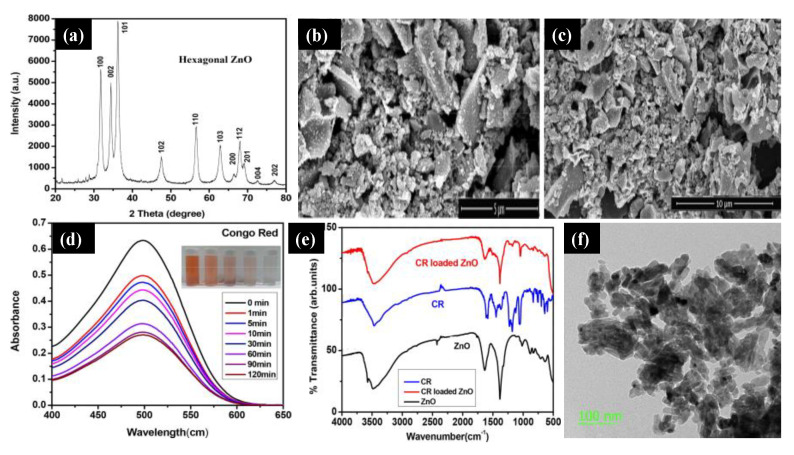
(**a**) X-ray diffraction pattern of the ZnO NPs. (**b**,**c**) Scanning electron microscopy images of agglomerated ZnO NPs. (**d**) UV–Vis spectra of the Congo red (CR) dye versus time. (**e**) The FTIR spectra of the ZnO NPs, CR dye, and CR–loaded ZnO NPs. (**f**) Transmission electron microscopy of the ZnO NPs at a 100 nm scale. Figure adapted with permission from Reference [39]. Copyright 2020 Elsevier.

**Figure 4 molecules-26-02236-f004:**
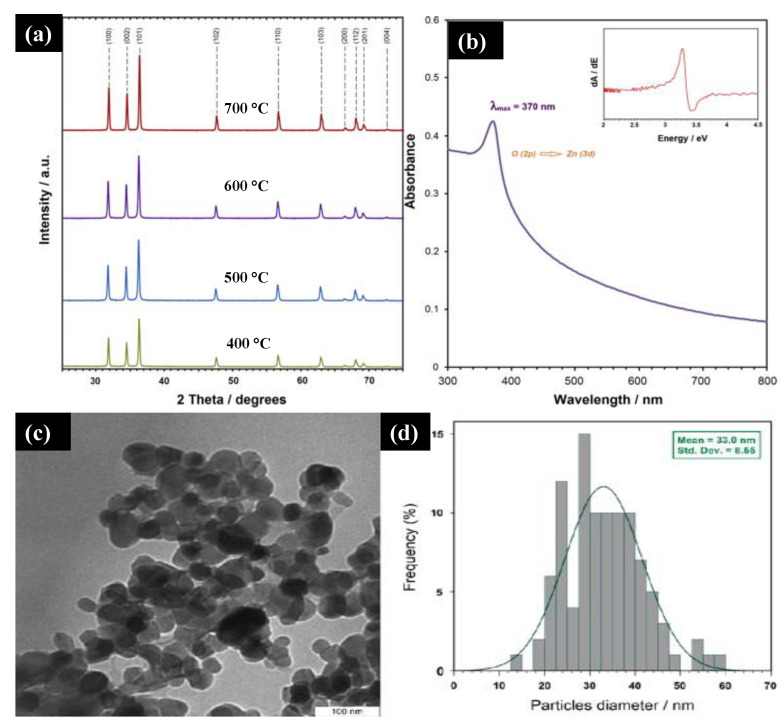
(**a**) XRD plot of synthesized ZnO NPs at different calcinations at 600 °C. (**b**) The UV-Vis spectra of the ZnO NPs synthesized at 600 °C. (**c**,**d**) TEM image and particle size distribution of ZnO NPs in gum tragacanth (GT) media synthesized at 600 °C. Figure adapted with permission from Reference [41] Copyright 2013 Elsevier.

**Figure 5 molecules-26-02236-f005:**
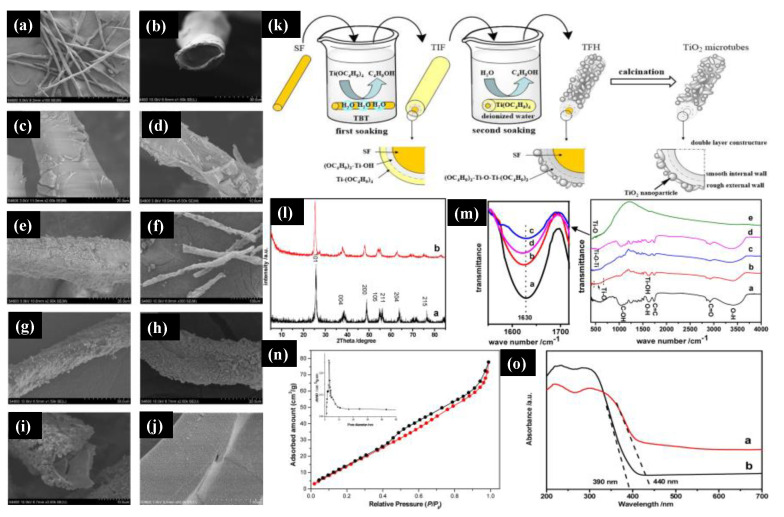
(**a**,**b**) Scanning electron microscopy images of the *Platanus acerifolia* seed fibers, (**c**,**d**) titanium sol-impregnated fibers (TIF), (**e**) titanium fiber hybrid (TFH), (**f**–**i**) hollow tubular TiO_2_ microtubes, and (**j**) an inner wall of the TiO_2_ microtubes. (**k**) Schematic of the double soaking sol-gel route for the preparation of TiO_2_ microtubes. (**l**) The XRD pattern of the TiO_2_ microtubes (**a**,**b**) their inner wall. (**m**) The FTIR spectra of the natural SFs (**a**) and TIF (**b**), TFH (**c**), and TiO_2_ microtubes (**d**) prepared from the natural SFs. (**n**) The BET surface is a pore-size distribution of the TiO_2_ microtubes. (**o**) UV-Vis spectra of the commercially available TiO_2_ (**a**) and synthesized TiO_2_ microtubes (**b**). Figure adapted with permission from Reference [89]. Copyright 2017 Elsevier.

**Figure 6 molecules-26-02236-f006:**
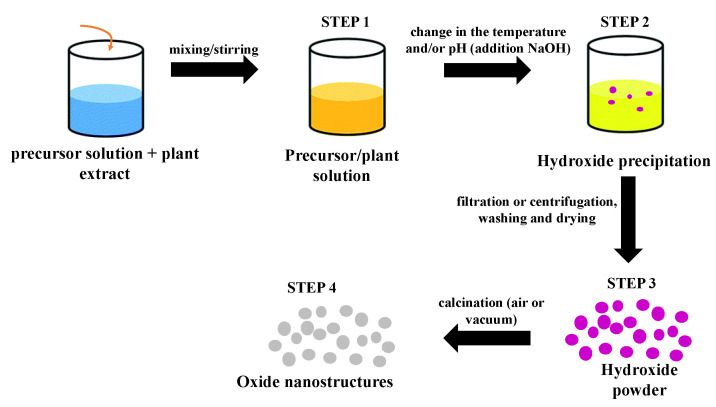
Schematic diagram showing all steps involved in a generic green synthesis mediated by plant extract using the co-precipitation method.

**Figure 7 molecules-26-02236-f007:**
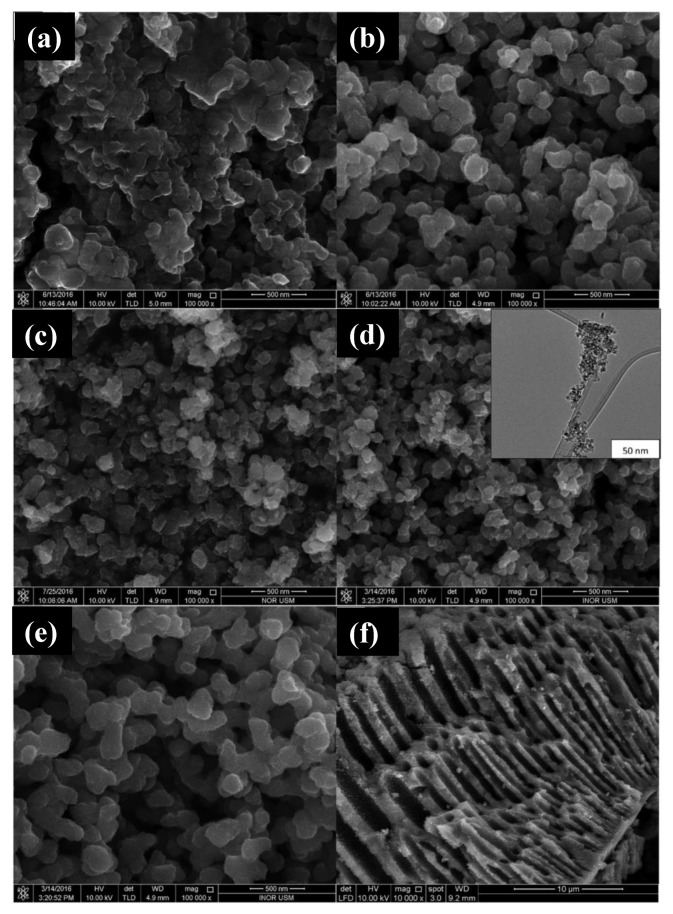
FESEM micrographs of TiO_2_ NPs synthesized using (**a**) uncalcined TiO_2_; (**b**) 0.01 mol titanium tetraisopropoxide (TTIP), pH 5; (**c**) 0.01 mol TTIP, pH 7; (**d**) 0.01 mol TTIP, pH 9, insert: HRTEM image; (**e**) 0.07 mol of TTIP in pH 9; (**f**) pore channels of TiO_2_ NPs. Figure adapted with permission from Reference [110]. Copyright 2017 RSC.

**Figure 8 molecules-26-02236-f008:**
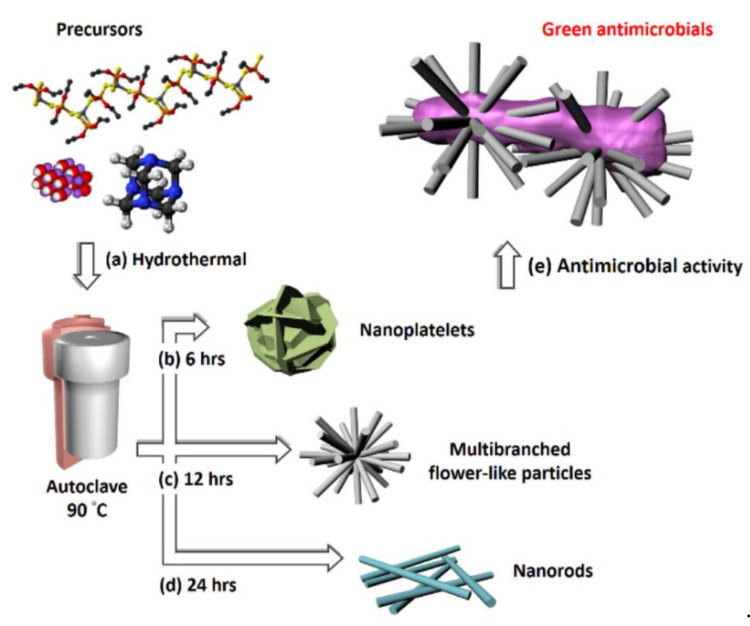
(**a**) Precursors (zinc acetate, hexamethylenetetramine (HMT), and sodium hydroxide) used in the synthesis of different nZnO morphologies; auto-clave treatment at 90 °C was maintained for (**b**) 6 h to get nanoplatelets, (**c**) 12 h to achieve multibranched flower-like particles, and (**d**) 24 h to obtain nanorods. (**e**) Diagram of the antimicrobial activity test for the ZnO multibranched flower-like particles. Figure adapted with permission from Reference [129]. Copyright 2020 Elsevier.

**Figure 9 molecules-26-02236-f009:**
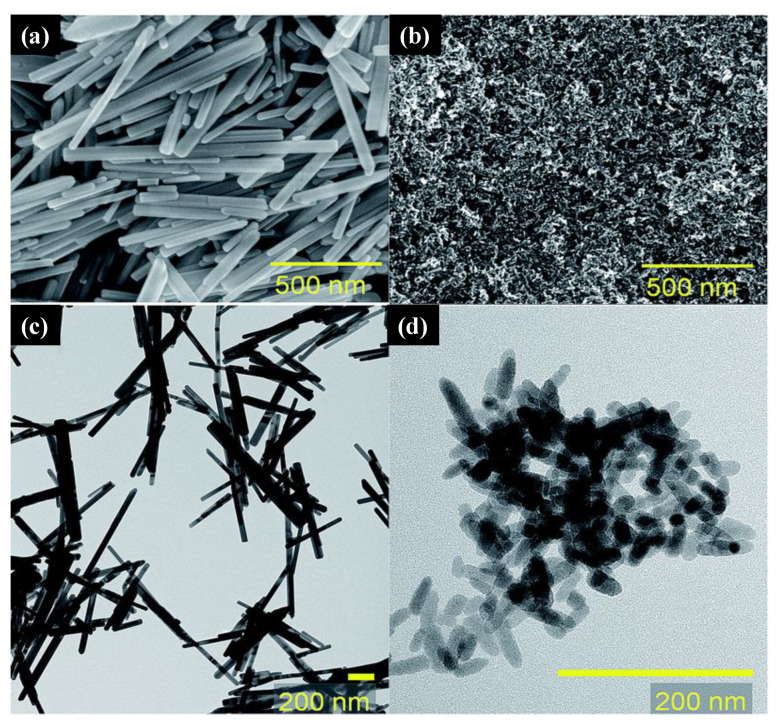
SEM images for ZnO (**a**) and Zn_0.85_Al_0.15_O (**b**) and TEM for ZnO (**c**) and Zn_0.85_Al_0.15_O (**d**). Figure adapted with permission from Reference [160]. Copyright 2020 RSC.

**Figure 10 molecules-26-02236-f010:**
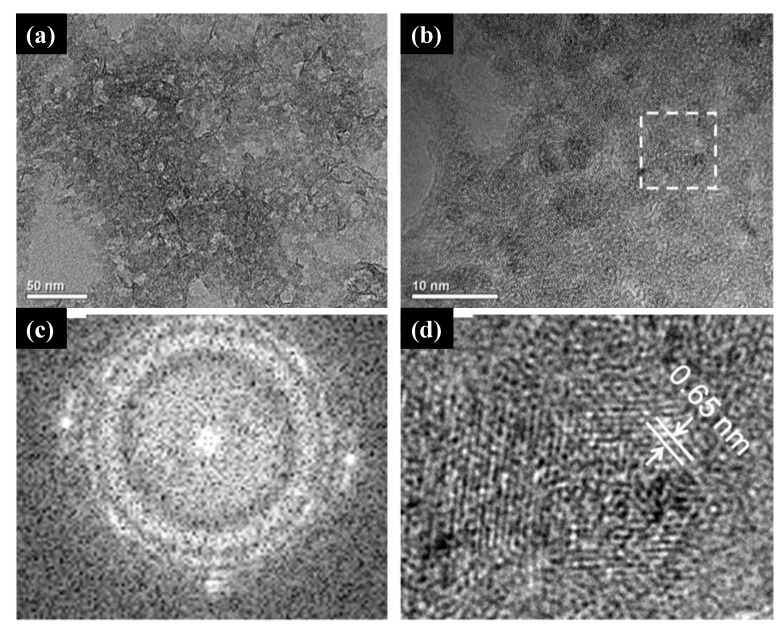
(**a**) TEM and (**b**) HRTEM image of the as-synthesized catalysts. (**c**) the fast Fourier transition (FFT) image of the selected area, and (**d**) inverse fast Fourier transition (IFFT) image obtained from (**c**). Figure adapted with permission from Reference [164]. Copyright 2010 Elsevier.

**Figure 11 molecules-26-02236-f011:**
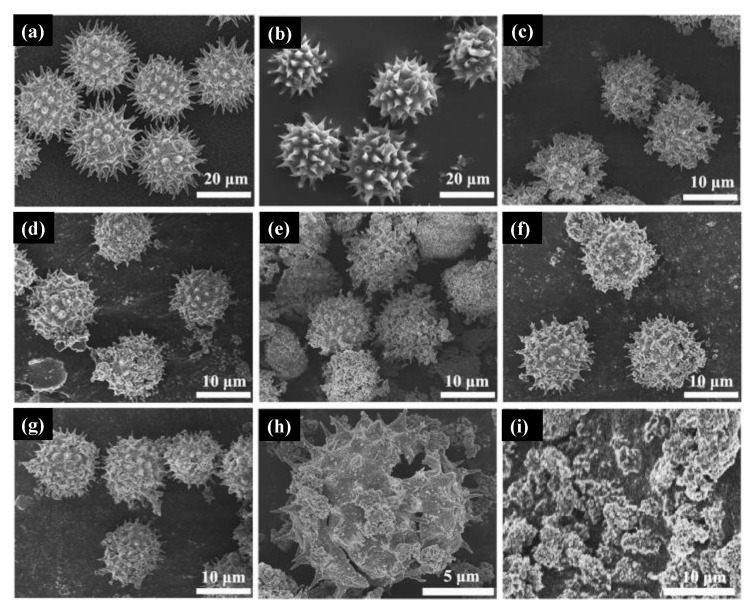
Micrograph of sunflower pollen (**a**), ZrO_2_-doped TiO_2_ spinous hollow sample before calcination (**b**), 4.6% doped TiO_2_ sample (**c**), 8.8% doped TiO_2_ sample (**d**), 12.6% doped TiO_2_ sample (**e**), 18.2% doped TiO_2_ sample (**f**), calcinated sample presented in a smaller size due to the removal of the pollen template (**g**), hollow microsphere (**h**), and TiO_2_ samples synthesized without templates (**i**). Figure adapted with permission from [167] Copyright 2018 Elsevier.

**Table 1 molecules-26-02236-t001:** Classification of five different types of “gels” essential to a material’s sol-gel synthesis [35].

Type of Gel	Bonding	Source	Gel Schematic
Colloidal	Particles connected by Van der Waals or hydrogen bonding	Metal oxides or hydroxide sols	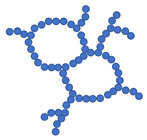
Metal-oxane polymer	Inorganic polymers connected via covalent or intermolecular bonding	Hydrolysis or condensation of metal alkoxides, e.g., SiO_2_ from tetramethyl orthosilicate	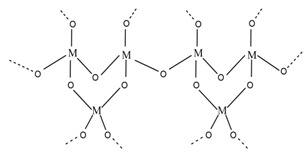
Metal complex	Weakly interconnected metal complexes	Concentrated metal complex solution, e.g., aqueous metal citrate or ethanolic metal urea often form resins or glassy solids rather than gels	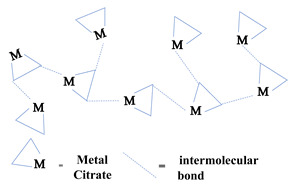
Polymer complex I in situ polymerizable complex (“Pechini” method)	Organic polymers interconnected by covalent or coordinate bonding	Polyesterification between polyhydroxy alcohol (e.g., ethylene glycol) and carboxylic acid with metal complex (e.g., metal-citrate)	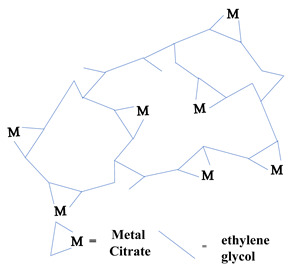
Polymer complex II coordinating and crosslinking polymers	Organic polymers interconnected by coordinate and intermolecular bonding	Coordinating polymer (e.g., alginate) and metal salt solution (typically aqueous)	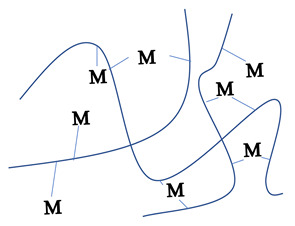

**Table 2 molecules-26-02236-t002:** Biosynthesis approaches to synthesize ZnO NPs from different plant sources.

Morphology/Shape	Plant Source	Zinc Precursor	Reference
Quasi-spherical	*Agathosma betulina*	Zinc nitrate hexahydrate	[43]
Hexagonal	*Allium sativum*, *Allium cepa*, and *Petroselinum crispum*	Zinc nitrate hexahydrate	[44]
Spherical	*Aloe barbadensis miller*	Zinc nitrate solution	[45]
Spherical, oval, and hexagonal	*Aloe vera*	ZnSO_4_	[46]
Not Reported	*Aloe vera*	Zinc nitrate, sodium hydroxide	[47]
Hexagonal	*Anchusa italic*	Zinc acetate dehydrate	[48]
Hexagonal	*Anisochilus carnosus*	Zinc nitrate hexahydrate	[49]
Spherical	*Artocarpus gomezianus*	Zinc nitrate hexahydrate	[50]
Spherical	*Aspalathus linearis*	ZnNO3, ZnCl2, and Zn-ammonium hydrate	[51]
Spherical	*Azadirachta indica*	Zinc nitrate hexahydrate	[52]
Various morphologies	*Azadirachta indica*	Zinc nitrate hexahydrate	[53]
Rod-shaped	*Black tea*	Zinc acetate dehydrate	[54]
Spherical	*Boswellia ovalifoliolata*	Zinc nitrate solution	[55]
Spherical and granular	*Calotropis procera*	Zinc acetate dehydrate	[56]
Hexagonal	*Caralluma fimbriata*	Zinc nitrate hexahydrate	[57]
Nano-flowers	*Carica papaya*	Zinc nitrate	[58]
Hexagonal	*Carica papaya*	Zinc nitrate solution	[59]
Flower-shaped NPs	*Carissa edulis*	Zinc nitrate hexahydrate	[60]
Spherical	*Cassia fistula*	Zinc nitrate hexahydrate	[61]
Spherical	*Citrus aurantifolia*	Zinc acetate dehydrate	[62]
Pyramid-like	*Citrus aurantifolia*	Zinc acetate dehydrate	[63]
Hexagonal	*Coffee*	Zinc acetate dehydrate	[64]
Polyhedron	*Corymbia citriodora*	Zinc nitrate solution	[65]
NPs	*Heritiera fomes* and *Sonneratia apetala*	Zinc chloride	[66]
NPs	*Jacaranda mimosifolia*	Zinc gluconate hydrate	[67]
Spherical and hexagonal	*L. leschenaultiana*	Zinc acetate dehydrate	[68]
Spherical	*Limonia acidissima* L.	Zinc nitrate solution	[69]
NR	*Mimosa pudica*	Zinc acetate dehydrate	[64]
Needle-like	*Nephelium lappaceum*	Zinc nitrate hexahydrate	[70]
Spherical	*Nephelium lappaceum* L.	Zinc nitrate hexahydrate	[71]
Hexagonal	*Ocimum basilicum* L. var. purpurascens	Zinc nitrate hexahydrate	[72]
Spherical	*Parthenium hysterophorous*	Zinc nitrate solution	[73]
Spherical and hexagonal	*Parthenium hysterophorus* L.	Zinc nitrate solution	[74]
Spherical	*Phyllanthus niruri*	Zinc nitrate solution	[75]
Triangular	*Physalis alkekengi* L.	Zinc contaminated soil	[76]
Spherical and hexagonal	*Plectranthus amboinicus*	Zinc nitrate solution	[77]
Rod-shaped	*Plectranthus amboinicus*	Zinc nitrate hexahydrate	[78]
Spherical	*Polygala tenuifolia*	Zinc nitrate hexahydrate	[79]
Spherical	*Pongamia pinnata*	Zinc nitrate hexahydrate	[80]
Spherical	*Rosa canina*	Zinc nitrate solution	[81]
Columnar	*Sedum alfredii*	ZnSO_4_	[82]
Hexagonal	*Solanum nigrum*	Zinc nitrate solution	[83]
Spherical	*Terminalia chebula*	Zinc nitrate hexahydrate	[84]
Spherical	*Tribulus terrestris*	Zinc oxide powder	[85]
Not Reported	*Trifolium pratense*	ZnO powder	[86]
Spherical	*Vitex negundo*	Zinc nitrate hexahydrate	[87]
Spherical	*Vitex trifolia* L.	Zinc nitrate hexahydrate	[88]

**Table 3 molecules-26-02236-t003:** Green sol-gel synthesis approach for the preparation of TiO_2_ nanostructures.

Material	Green Synthesis	Reactant	Reference
TiO_2_	*Lagenaria siceraria* and *Pithecellobium dulce leaf*	Titanium tetraisopropoxide, isopropanol, acetic acid, and ethanol	[92]
TiO_2_	*A. altissima* leaf extracts	Titanic acid and water	[93]
TiO_2_	*Leaf extract of L. siceraria*	Titanium (IV)-isopropoxide, ammonia, glacial acetic acid, and ethanol	[94]
TiO_2_	*Jatropha curcas* L.	TiCl_4_, ammonia	[95]
TiO_2_	*Acanthophyllum laxiusculum SchimanCzeika roots*	Titanium tetraisopropoxide(TTIP), 2-propanol, nitric acid	[96]
TiO_2_	*Pista Shell, Tamarind Seed, Corn Pith*	Isopropanol, titanium tetraisopropoxide, acetic acid (2%)	[97]
TiO_2_	*Green tea extract powder*	Titanium isopropoxide, isopropanol	[98]

**Table 4 molecules-26-02236-t004:** Green co-precipitation syntheses of ZnO nanostructures mediated by plant extract.

Morphology	Zinc Precursor	Plant/Part Used	Role of Biocomponents	Reference
Spherical NPs	Zinc acetate dihydrate	*Azadirachta indica*/leaf	Reducing and stabilizing agent	[111]
Flower-like structures	Zinc acetate dehydrate	*Laurus nobilis*/leaf	Reducing and capping agent	[112]
Quasi-spherical NPs	Hydrated zinc nitrate	*Agathosma betulina*/leaf	Oxidizing/reducing chemical agent	[43]
Spherical NPs	Zinc nitrate hexahydrate	*Tabernaemontana divaricata*/leaf	Capping and chelating agents	[113]
Spherical NPs	Zinc acetate dihydrate	*Carica papaya*/leaf	Capping and reducing agent	[114]
Spherical NPs	Zinc nitrate hexahydrate	*NepheliumLappaceum* L./fruit	Natural ligation agent	[71]
Nanoflowers	Zinc chloride	*Typha latifolia.* L./leaf	Reduction agent	[115]
Flower-like structure	Zinc acetate	*Kalopanax septemlobus/barks*	Reducing and capping agent	[116]
Rod-like and spherical NPs	Zinc nitrate	*Bambusa vulgaris and Artabotrys hexapetalu*/leaf	Reducing agent	[117]
Flower-like structure, cauliflower-like, and nanoflowers	Zinc nitrate hexahydrate	*Zea mays, Artocarpus heterophyllus, Punica granatum/husk, peel and peel*	Capping agent	[118]
Flower-like nanostructures	Zinc acetate	*Cyanometra ramiflora*/leaf	Reducing agent	[119]
Spherical NPs	Hydrated zinc chloride	*Broccoli*/leaf	Capping agent	[120]
Nanoflowers	Zinc acetate	*Citrullus lanatus/rind*	Reducing agent	[121]
Tetrameric structured NPs	Zinc nitrate hexahydrate	*Amomum longiligulare/fruit*	Reducing and stabilizing agent	[122]
Hexagonal NPs	Zinc nitrate tetrahydrate	*Andrographis paniculate*/leaf	Reducing agent	[123]
Leaf-like nanostructures	Zinc nitrate	*Rubus coreanus/fruit*	Reducing and capping agent	[124]

**Table 5 molecules-26-02236-t005:** Parameters involved in the process of dye photodegradation by TiO_2_ and ZnO nanostructures synthesized by green routes. The dyes methylene blue, rhodamine B, malachite green, methyl orange, Congo red and crystal violet are indicated by the abbreviations MB, RhB, MG, MO, CR and CV, respectively.

Morphology/Material Phase	Green Synthesis Method	Radiation	Dye	Dye Concentration	Catalyst Concentration	pH	Exposure Time (min)	Efficiency (%)	Reference
Nanorods/TiO_2_ anatase	Microwave	Artificial sunlight	RhB	10^−5^ M (50 mL)	50 mg/10 mL	dye pH	120	>98%	[182]
Spherical NPs/TiO_2_ anatase	Co-precipitation *	Solar light	Coralline red	5 mg/100 mL	10 mg/100 mL	8	140	92.17%	[103]
Meso/macro-porous nanostructures	Precipitation *	Sunlight	MB	20 mg/L	-	dye pH	135	>95%	[183]
Spherical NPs/TiO_2_ anatase	Continuous ultrasonic stimulation	UV light	MB	10 ppm	1 g/L	dye pH	150	92.5%	[184]
Elliptical NPs/TiO_2_ anatase	Sol-gel *	Visible light	MB, fuchsine, CV, and Rhodamine 6G	10 mg/L (100 mL)	0.1 g	dye pH	180	88–99%	[185]
Dandelion-like structures/TiO_2_ anatase-rutile	Hydrothermal	UV light	MB	10 mg/L (40 mL)	20 mg–40 mL	dye pH	650	>97%	[132]
Spherical NPs/TiO_2_ rutile	Microwave *	Sunlight	MB, MO, CV, and alizarin red	1 mg/100 mL	10 mg/50 mL	dye pH	360	77.3–92.5%	[186]
Spherical NPs/TiO_2_ anatase	Co-precipitation *	UV light	Reactive Green-19	6.7 mM	0.030 g/100 mL	3.5, 10.5	120	98.88%	[94]
Non-spherical NPs/TiO_2_ anatase	Sol-gel	UV light	MO	20 ppm (100 mL)	0.1 g	dye pH	150	94%	[187]
Spherical structures/TiO_2_ anatase	Precipitation	Sunlight	MB	6–40 ppm (200 mL)	0.05–0.40 g	dye pH	120	100%	[110]
Nanoflowers/ZnO wurtzite	Co-precipitation *	UV light	MB, MG, CR, and Eosin Y	15 mg/L	5 mg/L	dye pH	90	100%	[188]
Spherical and hexagonal prismatic NPs and nanosheets/ZnO wurtzite	Co-precipitation	Visible light	RhB	5 × 10^−6^ M (2 mL)	1 mg/2 mL	dye pH	120	75–84%	[125]
Leaf-like structures/ZnO wurtzite	Co-precipitation *	Dark condition	MG	10 mg/L (90 mL)	5 mg/90 mL	dye pH	240	~80%	[124]
Hollow microspheres/ZnO wurtzite	Hydrothermal *	UV light	MG	10 mg/L (200 mL)	1 g/L	5	60	~90%	[189]
Nanosheets/ZnO wurtzite	Hydrothermal	UV light	MB	1 × 10^−5^ M (200 mL)	0.05 g/200 mL	dye pH	50	99.2%	[190]
Flower-like nanostructures/ZnO wurtzite	Co-precipitation *	UV light	MB	50 µM	0.5–1.0 g/ml	dye pH	30	97.5%	[116]
Quasi-hexagonal NPs/ZnO wurtzite	Microwave *	UV light	MB	5 mg/L (100 mL)	30 mg (100 mL)	3–11	40	70–100%	[191]
Spherical NPs/ZnO wurtzite	Mechanically assisted metathesis reaction	UV light	MB	10 mg/L (100 mL)	10 mg/100 mL	dye pH	120	78%	[192]
Hollow nanospheres/ZnO wurtzite	Hydrothermal	UV and visible light	CR	20 ppm (50 mL)	25 mg/50 mL	5–9	90	99%	[193]
Spongy cave-like structures/ZnO wurtzite	Solution combustion *	UV and sun light	MB	5 ppm (100 mL)	50 mg/100 mL	2–12	90	~18–100%	[194]
Mysorepak-like, canine teeth, hollow pyramid, and aggregated hexagonal/ZnO wurtzite	Combustion *	UV light	MB	5–20 ppm (100 mL)	50–200 mg/100 mL	2–12	150	85–100%	[195]
Quasi-spherical NPs/ZnO wurtzite	Co-precipitation *	Sunlight	MB	1 × 10^−5^ M (100 mL)	100 mg/100 mL	dye pH	90	100%	[113]
Spherical NPs/ZnO wurtzite	Sol-gel	Visible light	Direct blue 129	20 mg/L (50 mL)	30–60 mg/50 mL	dye pH	105	~60–95%	[196]
Spherical NPs/ZnO wurtzite	Hydrothermal *	UV light	MB and MO	10 mg/L (50 mL)	1–30 mg/50 mL	dye pH	50–60	96.6–98.2%	[197]
Spherical and rod-like NPs/ZnO wurtzite	Co-precipitation *	Visible light	RhB	10 mg/L	1 g	dye pH	180	88–92%	[117]
Sponge-like structures/ZnO wurtzite	Combustion *	UV and sun light	MB and MG	5–25 ppm (100 mL)	50–200 mg/100 mL	2–12	120–150	~10–100%	[198]
NPs/ZnO wurtzite	Combustion *	UV light	Rose Bengal	2–40 ppm (250 mL)	20–80 mg/250 mL	6–10	90	~70–90%	[199]
Porous NPs/ZnO wurtzite	Solution combustion *	UV e sun light	MB	5–20 ppm (100 mL)	50–200 mg/100 mL	2–12	120	~3–99%	[50]
Spherical NPs/ZnO wurtzite	Combustion *	UV light	CR	10–40 ppm (250 mL)	20–80 mg/250 mL	6–10	60	70–90%	[200]
Hexagonal NPs/ZnO wurtzite	Solution combustion *	UV and sun light	MB	5–20 ppm (100 mL)	100 mg/100 mL	3–12	40–50	90–100%	[201]
Nanoflowers/ZnO wurtzite	Co-precipitation *	Sunlight	RhB	10 µM (100 mL)	20 mg/100 mL	dye pH	200	98%	[201]
Sphere-like nanostructures	Co-precipitation *	UV light	MB	50 µM	50 mg	dye pH	210	98.6%	[119]
Spherical NPs/ZnO wurtzite	Hydrothermal *	UV light	MB and MO	10 mg/L (50mL)	5–30 mg/50 mL	dye pH	50	96.6–98.2%	[202]
Spherical NPs/ZnO wurtzite	Sol-gel *	UV light	MB, MO, and Methyl red	5–25 ppm (50 mL)	50 mg/50 mL	dye pH	35	60–100%	[197]
Spherical morphology/ZnO wurtzite	Solvothermal *	Visible light	MB	20 mg/L (100 mL)	100 mg/100 mL	4.0–9.8	30	7.6–96.8%	[203]
Nanoflowers/ZnO wurtzite	Co-precipitation	Sunlight	Indigo carmine	-	50 mg	dye pH	120	83%	[204]
Quasi-spherical NPs/ZnO wurtzite	Combustion *	UV light	MB	5 × 10^−5^ M (30 mL)	20 mg/30 mL	5–12	120	40–96%	[115]
Plates, bullets, flower, prismatic tip, and closed pinecone nanostructures/ZnO wurtzite	Solution combustion *	UV and sun light	MB	10 ppm (250 mL)	60 mg/250 mL	dye pH	60	85–92%	[205]

The *** in the second column indicates that the green syntheses were aided by plant extracts.

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
