# Peer review of "Green Synthesis and Applications of ZnO and TiO2 Nanostructures"

_molecules, 2021, doi:10.3390/molecules26082236_

Round 1

Reviewer 1 Report

The submitted manuscript is a review report dealing with green technologies to synthesize ZnO and TiO2 nanostructures.  It was well organized and interesting for the researchers in this field.  The submitted manuscript can be accepted in the present form after minor revision.  My comments are as follows.

  1. Figure 3 (b) and (c) looks like expanded to longitudinal direction. It is difficult to see the scale bar on (f). Please check the original paper.
  2. Figure 4 (c) looks like expanded to lateral direction. Please check the original paper.
  3. There are miss typing in the text.

Author Response

Referee # 1

Comments: The submitted manuscript is a review report dealing with green technologies to synthesize ZnO and TiO2 nanostructures.  It was well organized and interesting for the researchers in this field.  The submitted manuscript can be accepted in the present form after minor revision.  My comments are as follows.

Comment [1]. Figure 3 (b) and (c) looks like expanded to longitudinal direction. It is difficult to see the scale bar on (f). Please check the original paper.

Response: Thank you for the suggestion. We have revised the figures with best efforts, however, we cannot edit original figures. The scale bar in Figure 3(f) is revised and visible in the revised manuscript.

Comment [2]. Figure 4 (c) looks like expanded to lateral direction. Please check the original paper.

Response: Thank you for the suggestion. We have revised the Figures with best efforts, however, the original figure’s quality is poor so we cannot edit.

Comment [3]. There are miss typing in the text.

Response: We have corrected them in the revised version.

Reviewer 2 Report

This review deals with the TiO2 and ZnO nanostructures prepared via green solvents and routes, and their applications.  A critical point of view is also offered. 

The review well describe the state of the art and include applications for the systems treated. The manuscript is well organized and does not result hard to read. English language is ok. 

For TiO2/ZnO application purpose, my suggestion is to integrate the relevant side regarding water splitting application. Here few references r which could help the authors: Chem. Rev. 1995, 95, 69-96; J. Phys. Chem. 1980, 84, 1987-1991; J. Phys. Chem. 1991, 95, 525-532; Int. J. Hydrogen Energy 40 2015 14483; Chem. Mater. 2002, 14, 4647–4653.

After this minor revision the manuscript could accepted for publications. 

Author Response

Referee # 2

Comments: This review deals with the TiO2 and ZnO nanostructures prepared via green solvents and routes, and their applications.  A critical point of view is also offered. The review well describes the state of the art and include applications for the systems treated. The manuscript is well organized and does not result hard to read. English language is ok.

Comment [1]. For TiO2/ZnO application purpose, my suggestion is to integrate the relevant side regarding water splitting application. Here few references r which could help the authors: Chem. Rev. 1995, 95, 69-96; J. Phys. Chem. 1980, 84, 1987-1991; J. Phys. Chem. 1991, 95, 525-532; Int. J. Hydrogen Energy 40 2015 14483; Chem. Mater. 2002, 14, 4647–4653.

Response: Thank you for the suggestion. We have added the water splitting application section (Section 3.5) in the revised manuscript. We have added the suggested references in this section as well.